# Cyberattacks in supply chains: A multi-case study

Xue Zhang[1], Xiaoya Ba[2], Bo Li[2,3]*

1 School of Business, Shenyang University of Technology, Liaoyang, Liaoning, China, 2 Ningbo China Institute for Supply Chain Innovation, Ningbo, Zhejiang, China, 3 School of Engineering, Massachusetts Institute of Technology, Cambridge, Massachusetts, United States of America

* libo@alum.mit.edu, libo@mit.edu

## Abstract

Supply chains are increasingly vulnerable to Supply Chain Cyberattacks (SCCAs) that exploit third-party trust and bypass traditional perimeter-based defenses. This study investigates the propagation mechanisms, impacts, and governance of SCCAs through a qualitative multi-case analysis of seven landmark incidents across diverse sectors, including retail, logistics, energy, and healthcare. Drawing on the Supply Chain Cyber Security System (SCCSS) framework, we map attack vectors, internal escalation pathways, and recovery dynamics across IT, organizational, and supply chain subsystems. Our cross-case synthesis reveals that SCCAs predominantly originate from third-party connections (contractual governance failures) and escalate through four recurring propagation mechanisms—Network Flattening, Alert Paralysis, Operational Coupling, and Relational Weaponization. The scale of disruption is systematically amplified by inter-system coordination failures, while resilience emerges only when proactive information sharing is activated by strong internal organizational readiness. We introduce the concept of synergy dependency, demonstrating that external relational governance is hierarchically contingent on internal organizational controls, and reconceptualize Points of Penetration (PoPs) as dynamic transmission mechanisms that convert localized digital breaches into systemic operational paralysis. This research offers empirically grounded insights that adapt the SCCSS framework from a classificatory tool into a process-oriented model capable of explaining how cyber risk propagates as a lifecycle of entry, transmission, and interruption. The findings contribute analytical interpretations to supply chain governance theory by showing that cyber resilience is conditionally interdependent across subsystems. Practically, the study offers actionable guidance for implementing secure architecture, cross-organizational threat intelligence sharing, and supplier-support programs to strengthen the resilience of complex global supply chain ecosystems.

**Data availability statement:** All relevant data is secondary and publicly available. The study's findings are derived from a qualitative multi-case analysis of seven documented cyberattacks. A comprehensive list of primary forensic reports, government directives, and investigative data sources is provided within the manuscript in Supporting information S1. No primary participant data was generated for this study. The full text of data sources is available at https://doi.org/10.5281/zenodo.19379859.

**Funding:** The author(s) received no specific funding for this work.

**Competing interests:** The authors have declared that no competing interests exist.

# 1 Introduction

Modern supply chains interconnect the global economy through complex, interdependent commercial relationships [1–3]. In this interconnected business environment, cyberattacks targeting supply chain enterprises have escalated by exploiting disparities in cybersecurity measures and inter-organizational dependence. Cyber threats now extend beyond internal IT infrastructure, as criminals increasingly leverage vulnerabilities in suppliers and service providers to penetrate well-defended systems. Supply Chain Cyberattacks (SCCAs) have become covert and complex, frequently bypassing traditional perimeter-based defenses.

Unlike traditional attacks, SCCAs exploit the trust and technological integration among business partners, often causing severe operational and financial damage as demonstrated by incidents like the Target data breach (2013), NotPetya (2017), and SolarWinds (2020). Decentralized supply chain governance and limited visibility complicates countermeasures, while existing strategies often prioritize technical defense over broader organizational resilience.

This study addresses these gaps by investigating the propagation of SCCAs and the impact of governance mechanisms. Adopting the Supply Chain Cyber Security System (SCCSS) framework, this study analyzes major cyber incidents to examine how coordination and information sharing shape resilience at the system level. Given the challenges in securing supply chains across technical, organizational, and human dimensions, we use the SCCSS framework to identify targeted mitigation strategies and conceptualize supply chain cyberattacks as a dynamic lifecycle of entry, escalation, and containment.

Research indicates that cyber vulnerability grows with the number of supply chain connections [4], as more intricate networks present more entry points for risk. A structured understanding of risks across all supply chain stages is therefore essential, leading to our first research question: **RQ1.** *How do failures in contractual governance and third-party oversight facilitate the initial entry of cyberattacks across diverse industries?* Furthermore, the danger of a data breach is magnified by insufficient cyber risk management; failures to manage internal dependencies effectively lead to the second question: **RQ2.** *In what ways does a deficit in internal governance and inter-system coordination exacerbate the propagation and operational impact of a cyber breach?* Finally, organizations must actively collaborate with trusted suppliers to protect their work and investments. This necessity for external cooperation informs our final question: **RQ3.** *How does relational governance, specifically proactive intelligence-sharing, influence the speed of containment and systemic recovery following an attack?*

This study offers analytical interpretations that inform supply chain governance theory by moving beyond a static, component-based view of cybersecurity toward a causal, system-level understanding of risk propagation. Specifically, we introduce the concept of synergy dependency, demonstrating that the three SCCSS subsystems—IT, organizational, and supply chain—are not merely interrelated but exhibit hierarchical dependence. That is, the effectiveness of external (relational) governance is contingent upon internal organizational readiness. This refinement shifts the

analytical focus from simple interaction effects to conditional dependencies that determine whether defensive capabilities translate into actual resilience.

In addition, this study reconceptualizes Points of Penetration (PoPs) as dynamic transmission mechanisms rather than static interfaces. By doing so, we extend the SCCSS framework from a classificatory tool into a process-oriented model capable of explaining how cyber disruptions propagate across interconnected systems. Building on cross-case analysis, we identify four recurring propagation mechanisms—Network Flattening, Alert Paralysis, Operational Coupling, and Relational Weaponization—which explain how localized breaches escalate into systemic failures. Together, these contributions bridge supply chain governance and cybersecurity research by offering a mechanism-based account of digital risk in complex, interdependent networks.

Beyond the analyzed cases, the findings of this study offer broader utility for systemic resilience across various industrial and regulatory domains. At a policy level, these results support the implementation of cross-sector regulatory standards that mandate relational obligations for proactive threat intelligence sharing (e.g., the U.S. Cybersecurity Information Sharing Act of 2015). Finally, the framework assists focal firms in designing support programs for peripheral suppliers, effectively mitigating the security investment asymmetries that attackers frequently exploit as paths of least resistance.

## 2 Literature review

### 2.1 Supply chain cyberattacks

Supply chain cyberattacks compromise organizations by exploiting trusted relationships with external partners to gain indirect access [5]. These hard-to-trace intrusions affect diverse industries and utilize various vectors. A primary category involves attacks on third-party software providers, where trust in legitimate updates is exploited to distribute malware—a method exemplified by the SolarWinds breach (implanted in Orion updates) and the Dragonfly/Energetic Bear campaign [6]. Other vectors include "Watering Hole" attacks that compromise frequently visited websites to infect visitors [7], breaches of data aggregators, and rare hardware tampering. The inherent multi-level structures of software supply chains introduce novel challenges in network security, where risk contagion mechanisms and complex causal relationships dictate the speed and severity of vulnerability propagation [8,9]. Overall, attackers target the weakest link in the trust chain, meaning a breach in a less secure partner can endanger the entire network [10].

To boost their defenses, companies focus on speed and partnering with cybersecurity specialists, in addition to problem escalation and supply chain risk management practices [11,12]. Moreover, Crosignani et al. [13] demonstrate that cyberattacks are not isolated events but can trigger cascading failures through supply chains, causing economic damage far exceeding the initial impact. The observed market response of supplier network restructuring based on cybersecurity reliability offers an empirical argument for a systematic study of SCCAs. Durugbo and Al-Balushi [14] conduct a multi-case study of supply chain crises and emphasize a customer focus to ensure resilience. With a system focus, Pérez-Morón [15] find that organization support and firm size facilitate the adoption of cybersecurity practices in Chinese supply chains, and Jeong et al. [16] analyze publicly reported cyber breaches and find that the focus of attackers shifts from employees to system vulnerabilities.

### 2.2 Risk and governance

Third-party cyber risk stems from external vendor flaws as highlighted by breaches like the Target incident [17]. Effective governance integrates technical and organizational protocols through essential practices: conducting pre-partnership due diligence, mandating security obligations via contracts or SLAs, and performing continuous monitoring and audits [18].

The transition toward highly digitized, interconnected supply chain environments necessitates a multidimensional approach to governance. Because modern supply chains function as complex adaptive systems, cyber vulnerabilities cannot be evaluated in isolated nodes; rather, dynamic risk assessment frameworks are required to map the cascading

 

interdependencies and mitigate the systemic uncertainties inherent in digitalized outsourcing [19]. Establishing effective supply chain governance operates across contractual, relational, and transactional dimensions to balance individual firm interests with network-wide interdependencies [20]. However, as supply chains increasingly adopt complex digital architectures, the novelty and structural complexity of these integrated technologies often exceed the protective limits of traditional contractual governance [21]. Consequently, safeguarding these digitalized multi-tier networks requires a shift toward relational governance mechanisms, where agility, continuous information exchange, and inter-organizational trust become essential to mitigating systemic uncertainties and risks [21]. Moreover, the effectiveness of cybersecurity outsourcing is heavily dependent on strategic organizational alignment [22,23]. Together, these perspectives underscore that building resilience against modern supply chain cyber threats requires governance models that are as dynamically integrated as the technological networks they seek to protect.

Additionally, governance requires risk-based supplier segmentation to strictly control high-access vendors and integrated business continuity plans for joint incident response [24]. This "trust but verify" approach extends enterprise security to partners, aligning with the SCCSS framework's emphasis on IT, organizational, and supply chain subsystems [25].

## 2.3 Research gaps and paper positioning

Several gaps emerge in literature. Firstly, there is a lack of qualitative frameworks integrating IT and supply chain subsystems, with existing knowledge relying on technical reports rather than comparative analysis of governance effectiveness. Secondly, a disconnect exists between supply chain risk management and cybersecurity; traditional research focuses on physical reliability, while cyber research often neglects inter-organizational aspects. Ghadge et al. [25] fills this theoretical gap by connecting trust and collaboration to network resilience.

Thirdly, governance mechanisms regarding the coordination of internal and external systems remain under-theorized and fragmented. Finally, few studies systematically analyze the interaction among IT, organizational, and supply chain systems.

This study adopts the Supply Chain Cyber Security System (SCCSS) framework [25] to analyze cyber threats through a multidimensional lens. These subsystems are connected by three primary points of penetration (PoPs): *technology, human resources, and physical processes*. This study bridges the aforementioned gaps using the SCCSS framework to provide cross-industry insights into inter-system coordination and resilience mechanisms.

While previous literature has frequently treated technical vulnerabilities and organizational resilience in isolation [26], this study examines the dynamic interactions among IT infrastructure, organizational practices, and external partnerships across diverse sectors. By employing a multi-case design to analyze seven landmark incidents, we address the fragmentation in current governance theories regarding the coordination of internal and external supply chain systems. Consequently, we respond to urgent calls for global governance strategies that secure supply chain networks and enhance resilience in a digitalized global economy.

## 2.4 Terminology

Based on Supply Chain Governance (SCG) theory, *robust governance of the supply chain system* involves coordinating rights, obligations, and resources via contractual and relational mechanisms to achieve supply chain goals [27]. Lu et al. [28] distinguish this "robust" ability as improving stability under disturbance through information processing, differing from post-event resilience. Tang [29] emphasizes robustness as pre-designed anti-interference capabilities. Operationally, this includes third-party risk management (TPRM) and other monitoring tools to maintain control during attacks.

Defined as managing dependencies among activities, the *inter-system coordination* concept within the SCCSS framework refers to aligning IT, organizational, and supply chain subsystems regarding access control and incident response [30]. Effective collaboration enhances overall response speed and prevention capabilities (NIST SP 800−161, Supply Chain Risk Management Practices for Federal Information Systems and Organizations).

*Supply-chain information sharing for cybersecurity* entails the proactive exchange of threat intelligence and vulnerability status within the network [31]. While NIST SP 800−161 outlines sharing entries, firms often utilize ISAC/ISAO structures for joint response [32]. Sharing enhances threat perception and risk prevention, measured by dimensions such as timeliness, coverage, executability, and institutionalization.

Robustness and resilience are complementary concepts. *Robustness* is the pre-designed capacity to withstand interference and remain stable, whereas *resilience* is the ability to recover and reconstruct post-impact [10]. Robust governance reduces attack propagation speed, while resilience mechanisms focus on mitigating losses and restoring operations. Assessment includes protective coverage for robustness and recovery time for resilience.

## 3 Research design and methodology

### 3.1 A multiple-case study approach

We adopt a multi-case design with three primary considerations. First, it enables cross-case comparison through literal and theoretical replication, which strengthens the robustness and analytical generalizability of our findings. Yin [33] suggests that each case in a multi-case study should be chosen to either predict similar results (literal replication) or predict contrasting results but for anticipatable reasons (theoretical replication) [34]. By analyzing multiple cases, the study can test whether similar governance factors produce consistent outcomes across different contexts [35]. Second, broadening the scope deliberately mitigates the selection bias inherent in studying exclusively high-profile, extreme cyber events. As noted in [36], multi-case analyses can discover technical and organizational risks in a variety of firms, ensuring that the identified vulnerabilities are systemic rather than anomalous. Third, given the issue's complexity, several cases fully expose the various elements of supply chain network vulnerabilities. Study [14] demonstrates the use of several examples in related research.

We investigate governance mechanisms using rich qualitative data. This approach advances theoretical progress given the exploratory nature of connecting supply chain problems with the SCCSS framework. After applying the rigorous methodology of independent study to each case, higher-level insights were extracted through cross-case synthesis.

### 3.2 Case selection, data collection, and analysis

The Supply Chain Cyber Security System (SCCSS) framework is in line with our deliberate case selection approach. To systematically address the universe of potential incidents, an initial candidate pool of 49 documented supply chain cyber-attacks occurring between 2010 and 2025 was aggregated (outlined in S2 List in S1 File). Rather than employing probability-based systematic sampling or convenience sampling, we utilized a highly structured purposive sampling strategy combining criterion and maximum variation sampling. Selection criteria required that each case involves a cyberattack targeting the SCCSS subsystems, offer rich verifiable data, and reflect diverse sectors and motives. Based on this selection criteria, seven cases were selected: Target Corporation, Maersk (NotPetya), SolarWinds, Colonial Pipeline, Quanta Computer, MediBank, and Toyota. These seven cases were deliberately chosen to satisfy replication logic, not sampling logic. Specifically, the cases represent both literal replication (where similar architectural vulnerabilities yield predictable, consistent failures) and theoretical replication (where contrasting architectures yield drastically different outcomes for anticipatable theoretical reasons). Finally, we acknowledge that focusing exclusively on high-profile, catastrophic breaches introduces an inherent risk of selection bias—specifically visibility bias, as successfully thwarted attacks rarely reach the public domain. However, this bias is deliberately mitigated through cross-sector maximum variation and data triangulation; the study leverages these extreme cases not to estimate statistical probabilities, but to map the mechanisms of complex systemic failure under extreme stress. As noted by [14], these diverse crises are intrinsically linked by their occurrence within and impact on the supply chain.

This study used a variety of public resources to guarantee rigor. Technical timeframes and behavioral insights were offered by investigative journalism, such as Greenberg's [37] report of NotPetya and Plachkinova and Maurer's [38] case

study on Target. The extent of the incident was corroborated by official government papers, such as CISA emergency instructions and forensic analyses from the US Senate [32]. These sources were supplemented by technical reports from firms like FireEye and Microsoft [39], alongside academic literature and corporate statements. Data collection followed an iterative process to construct event timelines analyzed against the three SCCSS subsystems. Given the reliance on secondary data, we employed data triangulation to validate the accuracy of the events. Technical details regarding attack vectors were sourced specifically from forensic reports (e.g., FireEye, Microsoft) and government filings (e.g., CISA, Senate investigations), while organizational impacts and timelines were cross-referenced with corporate press releases and investigative journalism. This multi-source approach mitigated the potential bias of any single report and ensured that the technical reality of each incident was accurate before analysis began.

To enhance analytical rigor and transparency, this study followed a structured qualitative analysis procedure consistent with established multi-case research methods [33,35]. The analysis utilized deductive thematic coding, where empirical observations were mapped to the SCCSS framework's definitions of points of penetration, risk propagation zones, and mitigation phases. For each case, evidence was extracted and coded according to the three SCCSS subsystems: IT Security, Organizational Security, and Supply Chain Security. Coding focused on (1) initial attack vectors, (2) internal propagation mechanisms, (3) governance or coordination failures, and (4) response and recovery actions.

To ensure the reliability of the qualitative analysis, the coding process followed a structured three-step protocol. First, a preliminary database was constructed for each case, aggregating timeline data from government reports, forensic analyses, and media coverage to ensure data triangulation. Second, *first-order codes* were assigned to specific events (e.g., stolen VPN password, delayed notification, manual system reboot). Third, these codes were mapped to the *second-order* theoretical categories of the SCCSS framework (IT, Organizational, or Supply Chain Subsystems). The codes are outlined in Table 1.

To ensure coding reliability and attenuate subjectivity, two researchers independently coded the empirical data across all seven cases. Following the initial independent coding phase, the researchers compared their assigned first-order codes and second-order theoretical categories. Any coding disagreements or discrepancies in categorization were systematically resolved through detailed discussion. During these reconciliation meetings, the researchers referenced the raw data and the established definitions within the SCCSS framework until a full consensus was reached. This collaborative and independent validation process ensured that the final themes were rigorously applied and remained consistent across the entire dataset.

In addition to theory-informed coding, we perform within-case analysis and cross-case pattern matching, using the Supply Chain Cyber Security System (SCCSS) framework as an analytical lens. Cases were systematically compared

**Table 1. Coding structure and data mapping.**

| Case | Raw Data Excerpt (Evidence) | First-Order Code | Second-Order Theme (SCCSS Subsystem) |
|---|---|---|---|
| **Target** | "Attackers stole the vendor's login credentials... HVAC service provider with remote access." | Third-Party Credential Compromise | **Supply Chain Security** |
| **Maersk** | "Global IT infrastructure... interconnected... lack of effective patch management and segmentation." | Flat Network Architecture/ Lack of Segmentation | **IT Security** |
| **SolarWinds** | "Vulnerability remained undiscovered for over a year... security team failed to identify abnormal traffic." | Delayed Detection/ Dwell Time | **Organizational Security** |
| **Toyota** | "Extreme reliance on the JIT model meant there was no buffer stock to absorb the disruption." | Lack of Redundancy/ JIT Fragility | **Supply Chain Security** |
| **Colonial** | "Lack of multi-factor authentication (MFA) and incident response planning." | Absence of Access Controls | **Organizational Security** |

to identify recurring configurations of subsystem weaknesses and governance failures. Patterns were considered robust when similar mechanisms appeared across multiple industries and attack types (e.g., third-party credential compromise combined with weak internal segmentation). This process enabled analytical generalization rather than statistical generalization.

Finally, we refine theoretical propositions through iterative comparison between the observed patterns and the SCCSS framework. Propositions were treated as analytical interpretations grounded in cross-case evidence, not as hypotheses subjected to statistical testing. We also contrast cases where similar attack vectors resulted in different outcomes, helping to identify moderating factors such as information-sharing speed and inter-system coordination.

This structured analytical process increases transparency, reduces narrative bias, and supports the credibility and replicability of the findings, consistent with qualitative research standards in supply chain and information systems research.

## 4 Theoretical framework

### 4.1 The supply chain cyber security system (SCCSS) framework

We utilize the Supply Chain Cyber Security System (SCCSS) framework [25] to classify cyber risks into interrelated information technology, organizational, and supply chain subsystems. Rooted in earlier supply chain risk management [40] and cross-organizational information security research [41], the SCCSS framework posits that security failures often stem from poor coordination across these systems, not from isolated vulnerabilities. This is because robust technology architectures must be inherently adaptive, meaning that static security perimeters are fundamentally insufficient without dynamic technological coordination across inter-organizational boundaries [42]. Consequently, we employ the SCCSS model to map complex attacks across IT infrastructure, organizational practices, and external partnerships [5]. As shown in Table 2, the SCCSS architecture comprises three interconnected governance subsystems.

This framework was selected to comprehensively assess technical vulnerabilities, organizational capabilities, and external interactions simultaneously. It facilitates the study of system coordination and security investment impacts, while supporting structured cross-case comparisons to understand governance challenges across industries.

The SCCSS framework posits that the three governance subsystems do not operate in isolation but are interconnected through three specific PoPs: *technology, human resources, and physical processes*. These PoPs serve as the operational "connective tissue" where inter-system coordination either facilitates resilience or allows for risk propagation. The *Technology PoP* acts as the digital interface between IT and supply chain subsystems, encompassing shared protocols and automated systems where a failure in one node can pause both subsystems. The *Human Resources PoP* bridges the Organizational and IT subsystems, representing the workforce's capacity to interpret and act upon technical data; failures here are exemplified by alert fatigue or the inability of security teams to execute timely interventions despite receiving automated system warnings. Finally, the *Physical Processes PoP* links the Supply Chain and Organizational subsystems

**Table 2. SCCSS framework – three subsystems overview.**

| Subsystem | Key Focus Areas | Description |
|---|---|---|
| Information Technology Security | Data asset protection; vulnerability management; network architecture and access control | focuses on using encryption, access control, patching, antivirus software, and IDS/IPS to protect internal IT infrastructure, including servers, endpoints, and data. |
| Organizational Security | Security governance mechanisms; staff training and awareness; incident response and recovery | Emphasizes interdepartmental collaboration, cybersecurity awareness, internal governance, decision-making processes, and incident response preparation. |
| Supply Chain Security | Third-party risk management; information sharing and collaboration; contractual and audit mechanisms | Focuses on security that is external to the supply chain, including vendor screening, joint audits, contractual security provisions, and cooperative threat detection. |

through real-world workflows and logistical dependencies, such as the "Just-in-Time" (JIT) manufacturing model. Resilience is manifested at this point by "Just-in-Case" contingencies, such as the manual paper-based recovery procedures utilized by Maersk to maintain baseline operations during a total digital paralysis. By focusing on these PoPs, the framework moves beyond a static view of security to a dynamic model where digital resilience is determined by the structural coordination across these overlapping operational zones.

Under SCCSS, risks are classified by the affected system:

- **IT System Risks:** Vulnerabilities in technical infrastructure include out-of-date patches, lax access control, and incorrect configuration.

- **Organizational System Risk:** Inadequate management commitment, staff knowledge, policy, or event response readiness.

- **Supply Chain System Risks:** Risks introduced by outside vendors, mistakes in information exchange, and inadequate external cooperation.

Applying SCCSS, this study conducts a qualitative analysis of multi-industry cases to evaluate subsystem responses. This approach reveals how system weaknesses influence attack vectors.

### 4.2 Conceptual model and propositional development

Our study advances six theoretical propositions to guide case analysis.

**Proposition 1 (Contractual Governance and Risk Incidence):** Mature supply chain contractual governance mechanisms, such as strict vendor due diligence and continuous monitoring, reduces the incidence of third-party risks and enhances effective containment. By establishing clear security obligations and audits, these formal structures act as anti-interference capabilities. They prevent attackers from exploiting the weakest link in the trust chain, addressing the fundamental oversight failures that facilitated the SolarWinds and Target breaches.

**Proposition 2 (Internal Governance and Lateral Spread):** Strong internal controls within IT and Organizational subsystems limit lateral spread and impact of a breach. Internal governance requires network segmentation and zero-trust access protocols. This ensures that a compromise at a peripheral, less-secure vendor does not cascade into core operational paralysis, directly mitigating the architectural vulnerabilities exposed in the Maersk and Colonial Pipeline attacks.

**Proposition 3 (Relational Governance and Threat Mitigation):** Relational governance mechanisms, specifically timely and executable information sharing, shortens attack dwell time and accelerates joint response. Relying on inter-organizational trust and continuous communication, proactive intelligence exchange builds network-wide threat perception. As demonstrated by FireEye's rapid disclosure during the SolarWinds incident, this relational agility significantly reduces the window of global exposure.

**Proposition 4 (Governance Synergy and Actionability):** Inter-system coordination determines whether information sharing translates into effective action. The mere availability of shared external intelligence is fundamentally insufficient without the internal organizational agility to execute a coordinated defense. This synergy gap explains why the availability of external threat data failed to trigger timely internal intervention during the MediBank breach, highlighting that relational trust must be paired with operational readiness.

**Proposition 5 (Operational Coupling and Disruption Velocity):** Tight operational coupling without corresponding digital redundancy amplifies the physical disruption velocity of a cyberattack across the supply chain. Building on the structural dependency of Supply Chain Governance theory, this dynamic is illustrated by the Toyota incident. Lean manufacturing creates rigid digital dependencies where a single supplier's failure forces systemic shutdowns.

**Proposition 6 (Security Investment Asymmetry):** In decentralized supply chains, severe asymmetry in cybersecurity investments between focal firms and peripheral suppliers creates a predictable path of least resistance for attackers.

Leveraging governance incentive and control mechanisms, this proposition addresses breaches like those at Quanta Computer and Target.

## 5 Individual case analyses

This section analyzes seven landmark cases across diverse industries—including retail, logistics, and healthcare—using the SCCSS framework. Table 3 provides a comparison of these incidents.

To ensure a systematic and transparent comparison across the diverse industries studied, Table 4 operationalizes the core analytical constructs by mapping the specific points of penetration, internal propagation mechanisms, and recovery timelines observed in each incident.

### 5.1 Case A: Target Corporation data breach (Retail, 2013)

This section analyzes the 2013 Target data breach, a landmark event that highlighted the systemic risks of third-party access. To ensure the rigor of the analysis, the case reconstruction relies on a diverse set of authoritative sources: Plach-kinova and Maurer's detailed case study of the breach [38], The U.S. Senate Commerce Committee investigation [17], academic studies such as Pigni et al. [43], and official statements from Target Corporation [44].

Target Corporation experienced one of the most well-known SCCAs in history in 2013. An HVAC (heating, ventilation, and air conditioning) service provider was the source of the hack in late 2013. By taking advantage of the trust that exists in the supply chain connection (the vendor has remote access to Target's network), attackers gained access to Target's systems by stealing the vendor's login credentials [44]. Target had already implemented a $1.6 million malware detection tool to notify them of data breaches, so they were theoretically prepared for hacking attempts. But by the end of 2013, Target failed to respond to the attack promptly [45].

The case contains four attack paths and methods:

- Initial Infiltration: Attackers circumvented direct perimeter protections by gaining access to Target's network using the obtained HVAC vendor credentials.

- Malware Deployment: After gaining access, attackers infected Target's point-of-sale (POS) systems with malware to steal credit card information during transactions.

**Table 3. Overview of cases and their key attributes (chronological).**

| Case No. | Case | Industry Type | Attack Time | Attack Type | Brief Description |
|---|---|---|---|---|---|
| A | Target Corporation Data Breach | Retail | 2013 | Third-Party Access | Attackers infiltrated via HVAC vendor, accessed POS systems, and stole 40 million credit card records. |
| B | Maersk NotPetya Attack | Global Logistics | 2017 | Weak Network Segmentation | NotPetya malware spread internally due to lack of segmentation, halting all operations. |
| C | SolarWinds Supply Chain Attack | Software Development | 2020 | Compromised Update Channel | Malware embedded in Orion updates affected 18,000+ global customers. |
| D | Colonial Pipeline Ransomware | Energy Transport | 2021 | Weak Response Planning | Attackers encrypted systems; company paid ransom and shut down pipeline. |
| E | Quanta Ransomware Incident | Electronics Manufacturing | 2021 | Data Leakage to Clients | REvil group stole Apple product blueprints, exposing OEM data control issues. |
| F | MediBank Data Breach | Healthcare Insurance | 2022 | Poor Data Access Control | Attackers accessed sensitive patient data, triggering public outcry. |
| G | Toyota Tier-1 Supplier Attack | Automotive Manufacturing | 2022 | Vendor System Vulnerability | Kojima Industries breached, leading to shutdown of 14 Toyota factories in Japan. |

**Table 4. Cross-case operationalization of key governance and resilience constructs.**

| Case | Entry Vector (Point of Penetration) | Governance or Coordination Failure | Internal Propagation Mechanism | Detection Delay (Dwell Time) | Mitigation and Operational Response | Recovery & Impact Metrics | Inter-System Coordination Failure |
|---|---|---|---|---|---|---|---|
| Target | HVAC vendor credentials | IT alerts vs. Organizational response failure | Alert Paralysis: Ignored system warnings | Weeks (Late Nov. to mid-Dec.) | POS terminal malware detection and lawsuits | 40M credit cards stolen | IT alerts vs. Org. response |
| Maersk | Hacked M.E.Doc update | IT infrastructure vs. Supply Chain updates | Network Flattening: Global IT spread | Minutes (Near-instant) | Manual paper records and offline recovery | Two-week disruption; $250M+ loss | IT infra vs. Supply Chain updates |
| SolarWinds | Trojan Orion update | IT verification vs. Supply Chain trust gap | Alert Paralysis: Undetected movement | 15 Months | Display indicators of breach | 18,000 clients; 100 high-value targets | IT verification vs. SC trust |
| Colonial | Stolen VPN password | IT access vs. Organizational planning | Network Flattening: IT-to-OT risk spread | Immediate (Encryption) | TSA Emergency Directives and ransom recovery | 6-day halt; $4.4M ransom paid | IT access vs. Org. planning |
| Quanta | Network infiltration | Supply Chain data control vs. Reputation | Relational Weaponization: Dual extortion | Short (Product launch) | Enforcing strict data-sharing agreements | $50M ransom; stolen Apple blueprints | SC data control vs. Client reputation |
| MediBank | Contractor credentials | Organizational access vs. IT security | Relational Weaponization: Data exfiltration | Weeks (Ongoing theft) | Sanctions and class-action lawsuits | 9.7M records; 200 GB stolen | Org. access vs. IT security |
| Toyota | Supplier remote access | Supply Chain JIT vs. Organizational continuity | Operational Coupling: Kanban halt | 48 Hours (Notification delay) | Hardening Tier-1 and Tier-2 defenses | 13,000 vehicles; 14 plants shut | SC JIT vs. Org. continuity |

- Lateral Movement: Because Target had not successfully implemented network segmentation, the attackers were able to travel freely from the vendor-accessible network segment to systems holding sensitive consumer data.

- Detection Failure: The security team failed to promptly respond to the alarms that Target's security software produced regarding the suspicious activity.

The attack had a dual impact. It resulted in the significant data leakage of roughly 40 million credit card records and 70 million customer personal details, including names, addresses, and phone numbers [43]. Concurrently, it caused severe financial and reputational harm: Target suffered major losses, faced numerous lawsuits, and saw executive turnover, severely damaging its standing as a trusted retailer.

**Key Lessons and Insights (SCCSS Mapping):** The Target case demonstrates how a combination of failed third-party oversight, poor internal segmentation, and uncoordinated incident response can create systemic vulnerabilities.

- IT Security Subsystem: The case exposes the vulnerability of POS terminals to malware implantation due to a lack of endpoint detection.

- Organizational Security Subsystem: Inadequate configuration of supplier access controls, combined with a weak security culture, reduced internal audits to procedural formality. Critical alerts from security systems were generated but ignored due to organizational failure, dramatically increasing the severity of the breach.

- Supply Chain Security Subsystem: HVAC service providers, authorized to access core systems, underwent no cybersecurity review. This failure to vet and monitor high-access external vendors created a critical attack vector.

## 5.2 Case B: Maersk & NotPetya attack (Global Logistics, 2017)

This section examines the 2017 NotPetya attack on Maersk, a defining moment that revealed the fragility of global inter-connected supply chains. To ensure analytical rigor, this study draws on a diverse range of public sources. These include detailed investigative journalism, such as Andy Greenberg's [32] account of NotPetya; technical reports from cybersecurity firms like Microsoft [39]; and official corporate statements, such as Maersk's post-incident report [46], which help establish the incident's scope.

The NotPetya malware outbreak devastated the multinational logistics and shipping behemoth Maersk in June 2017. Initially posing as ransomware, NotPetya turned out to be a damaging wiper attack with connections to state-sponsored actors. By taking advantage of confidence in reputable business tools, the attack expanded via a hacked software supply chain [47].

There are three attack paths and methods in this case. First, the software was compromised. Updates for M.E.Doc, a well-known Ukrainian tax accounting program utilized by Maersk's Ukraine office, contained the malware. The malware entered Maersk's network after the update was implemented on its systems. Second, the organization and IT infrastructure allowed rapid propagation of the malware. NotPetya exploited unpatched Windows vulnerabilities and Maersk's flat, interconnected network to propagate across 600 sites in 130 countries within minutes. The malware then encrypted core systems, physically crippling operations by halting port activity, shipping coordination, and internal communications.

The attack had financial and operational consequences. On the one hand, staff depended on manual procedures like paper records and personal information applications to maintain the bare minimum of functionality during the roughly two-week disruption to Maersk's worldwide business. However, the corporation lost between $250 and $300 million in revenue.

**Key Lessons and Insights (SCCSS Mapping)** The Maersk case illustrates the catastrophic potential of a compromised software supply chain, demonstrating how a single, trusted third-party update can trigger global operational disruption.

- IT Security Subsystem: Paralysis of Active Directory, coupled with failed backups and absent redundancy, demonstrates the consequences of a non-segmented IT infrastructure and ineffective patch management.

- Organizational Security Subsystem: Issues included slow response to emergency procedures and employees' failure to isolate infection nodes in a timely manner.

- Supply Chain Security Subsystem: Malicious code was disseminated through the unverified M.E.Doc software from a Ukrainian supplier.

Nonetheless, Maersk responded to the worldwide "NotPetya" viral onslaught with readiness [47]. Maersk's recovery highlights the critical value of adaptable planning and offline recovery capabilities, offering vital lessons for logistics firms and multinational corporations facing similar systemic threats.

## 5.3 Case C: SolarWinds Orion supply chain attack (IT Software, 2020)

This section details the SolarWinds Orion attack, a sophisticated supply chain compromise discovered in late 2020 that leveraged trusted software updates to infiltrate high-value targets. Key evidence includes the CISA Emergency Directive on SolarWinds [32], the FireEye "Sunburst" malware analysis [48], and academic studies by Martinez and Duran [49]. These sources provide a comprehensive view of the technical mechanisms and the subsequent regulatory response.

One of the most advanced SCCAs to date is the SolarWinds attack, which was discovered in December 2020. The US IT management software company SolarWinds was the subject of an attack by a Russian government-funded advanced persistent threat (APT) group. Martinez and Duran noted that the attackers used malware to infect over 18,000 clients

from a variety of industries and over 40 public entities [49], including government agencies, retail businesses, and financial institutions.

The SolarWinds attack followed a four-stage path: initially, attackers infiltrated the SolarWinds development environment to insert the "Sunburst" malware into Orion software updates; these trojanized, digitally-signed updates were then distributed to about 18,000 customers via trusted channels, creating widespread backdoor access. Once inside, the stealthy malware imitated legitimate Orion activity to move laterally undetected, enabling attackers to focus on roughly 100 high-value targets, including U.S. agencies and Fortune 500 firms. The operation remained hidden for over 15 months—from September 2019 to December 2020—resulting in prolonged, large-scale espionage.

This cyberattack has had a wide-ranging and significant impact. First, its scope is broad, having breached numerous critical entities including key U.S. government departments—the Treasury, Commerce, and Homeland Security—as well as major corporations such as Microsoft and FireEye. Second, the attack has driven substantial policy responses, notably triggering a U.S. Executive Order focused on software supply chain security and leading to the creation of a Cyber Unified Coordination Group to manage recovery efforts.

**Key Lessons and Insights (SCCSS Mapping)** the SolarWinds incident revealed profound vulnerabilities within the software development lifecycle, particularly highlighting the dangers of excessive trust. In an ecosystem where a single, respected vendor is granted broad, uninspected access, a compromise at that source can propagate with devastating efficiency across countless networks, turning a trusted update into a powerful vector for widespread infiltration.

- Supply Chain Security Subsystem: This incident reveals that unverified code and a non-zero-trust supply chain are critical weaknesses. The solution requires mandatory internal verification and external transparency for all vendors.

- Organizational Security Subsystem: The security team failed to identify abnormal traffic in a timely manner.

- IT Security Subsystem: The root cause was a backdoor implanted within the Orion update package, coupled with a failure in detection mechanisms due to inadequate monitoring and weak anomaly identification.

This incident exploited blind trust in third-party software, forcing global organizations to reassess their IT supply chain integrity and tighten update controls. The swift sharing of IOCs was critical in containing the attack.

### 5.4 Case D: Colonial Pipeline ransomware attack (Energy Infrastructure, 2021)

This section examines the Colonial Pipeline ransomware attack, a watershed event that exposed the fragility of critical energy infrastructure to cyber extortion. Information of this case is drawn from the TSA Cybersecurity Directive [50], which marked a regulatory shift, alongside Reuters coverage of the ransom recovery by Wolf [51], and technical analyses of the attack vector by Mittal [52].

The DarkSide criminal group launched a ransomware attack against Colonial Pipeline, the operator of the biggest petroleum pipeline in the United States, in May 2021. The event exposed serious weaknesses in vital infrastructure supply lines and hindered fuel deliveries to the East Coast.

In this instance, there are three attack routes and techniques. First, hackers used stolen virtual private network (VPN) passwords to access the company's IT system, causing a great deal of chaos [52]. Forensic analysis suggests a compromised credentials database was the likely source of the exposed password. Second, the group encrypted data on Colonial's IT systems after gaining access. Third, Colonial stopped operations in advance out of concern that it would spread to the pipeline's operational technology (OT) systems.

There were significant financial losses associated with the attack. On the one hand, the closure caused price hikes, panic buying, and significant gasoline shortages on the East Coast [46]. However, Colonial spent about 4.4 million US dollars in Bitcoin to get a decryption key, some of which the FBI eventually found.

**Key Lessons and Insights (SCCSS Mapping)** The attack's scale and impact made it a watershed event, triggering a swift federal response, including TSA Emergency Directives mandating cybersecurity for pipeline operators [50].

- Organizational Security Subsystem: The successful breach points to deficiencies in two key areas: the absence of multi-factor authentication (MFA) and inadequate incident response planning.

- IT Security Subsystem: The leak of a single VPN credential crippled nearly half of the East Coast's fuel supply. This incident underscores the urgent need for stronger access controls, segmentation between IT and OT networks, and better incident response in the energy sector.

- Supply Chain Security Subsystem: The sector, while not the initial entry point, would have benefited from more integrated and timely information sharing across organizations once the attack was underway.

A single compromised password crippled nearly half the East Coast's fuel supply, exposing the critical need for robust access controls, strict IT/OT segmentation, and comprehensive incident response planning in critical infrastructure.

### 5.5 Case E: Quanta Computer REvil ransomware attack (Electronics Manufacturing, 2021)

This section analyzes the Quanta Computer ransomware attack, which introduced the "dual extortion" tactic targeting downstream clients through their suppliers. Key sources of this case include report by Newman [53] on the implications for Apple, and threat intelligence regarding the REvil leak site [54].

The REvil ransomware group attacked Taiwanese electronics manufacturer Quanta Computer, a significant supplier to Apple, in April 2021. The attack demonstrated a novel approach to cyber extortion by focusing on indirectly extorting Apple through its supply chain. Its client data is extremely useful for extortion as a contract manufacturer for many digital giants, and third-party suppliers with inadequate security investment may also serve as a springboard for additional attacks [53].

In this instance, there are two attack routes and strategies. Initially, REvil broke into Quanta's network, encrypted Quanta's computers, and took private Apple design schematics. Second, REvil promised to release the stolen data unless Apple paid a $50 million ransom, rather than focusing solely on Quanta. Snippets of the schematics were made public to put pressure on Apple during a product launch.

The attack has two effects. On the one hand, the hack revealed Apple's trade secrets, underscoring the dangers to intellectual property in production supply chains. However, despite not being physically hacked, Apple came under public criticism, demonstrating how supply chain flaws may hurt downstream brands.

**Key Lessons and Insights (SCCSS Mapping)** The Quanta Computer incident exposed how malicious actors can leverage trusted relationships and shared information in a supply chain to enable an attack.

- Supply Chain Security Subsystem: This incident exposes critical gaps, including supply chain vulnerabilities, the absence of OEM-level security audits, and an open-loop incident response that lacks closure.

- Organizational Security Subsystem: This case highlights systemic vulnerabilities that leave organizations unable to contain or respond to ransomware, exemplified by the growing trend of "Double Extortion"—where attackers steal data to coerce victims [55].

- IT Security Subsystem: Despite limited visibility into Quanta's internal security settings, the lack of adequate encryption and endpoint protection highlights critical vulnerabilities within its IT security framework.

This case serves as a critical warning that a single supplier's cybersecurity weakness can jeopardize the intellectual property and brand reputation of large clients like Apple. It underscores the need to extend security standards upstream, enforce strict data-sharing agreements, and strengthen protection for sensitive information across high-value supply chains.

### 5.6 Case F: MediBank data breach (Healthcare, 2022)

This section analyzes the MediBank data breach, a catastrophic event that exposed the sensitive health records of millions due to failures in third-party access management. Key evidence of this case study includes the Australian Information Commissioner's proceedings against MediBank [56], academic analysis of the psychiatric health ramifications by Looi et al. [57], and investigative reporting by ABC News Australia [58].

The largest health insurance company in Australia, MediBank, experienced a data breach in October 2022 that resulted in the disclosure of 9.7 million clients' private and medical information. Third-party risk in healthcare supply chains was highlighted by the attack, which was connected to the BlackCat/ALPHV ransomware syndicate [56].

In this instance, there are two attack routes and strategies. Third-party credential theft is the first. A contractor IT service desk employee provided high-level access credentials to the attackers. The attackers were able to get access to MediBank's network by using these credentials because the bank neglected to activate MFA on remote access VPNs. Significant data exfiltration is the second issue. The group stole 200 GB of data over the course of several weeks, including personal information and health claims records (diagnoses, procedures). The group posted portions of the data on the dark web after MediBank declined to pay the ransom.

Once again, the attack increased awareness of supply chain cyberattack dangers. On the one hand, there are dangers to one's safety and privacy. In the Medibank case, there was a data breach. The leak exposed patients and doctors to fraud, identity theft, and extortion—especially if sensitive mental health or substance use records were published [57]. MediBank also faced severe repercussions, including sanctions, multimillion-dollar fines, and class-action lawsuits.

**Key Lessons and Insights (SCCSS Mapping)** The MediBank vulnerability emphasizes the serious cybersecurity issues the healthcare sector faces, and attacks may primarily target third-party service providers with privileged access credentials. The shortcomings of the subsystems are reflected in this case.

- Supply Chain Security Subsystem: The outsourcing of medical services suffers from insufficient certification requirements, undefined security standards, and ambiguous supplier accountability.

- IT Security Subsystem: The lack of encryption in medical databases creates a critical vulnerability, enabling attackers to gain access via open or unprotected interfaces.

- Organizational Security Subsystem: Incident communication is delayed, and mandatory access authorization management has not been established. Widespread lateral movement and data breaches resulted from the organization's inadequate network fragmentation and failure to implement fundamental access controls like MFA.

The MediBank breach shows that decisions about third-party staff require a systems-wide risk perspective across governance, IT, and vendors, highlighting the need for strong internal controls and effective governance.

### 5.7 Case G: Toyota Tier-1 supplier cyber attack (Automotive Manufacturing, 2022)

This section analyzes the 2022 cyberattack on Kojima Industries, a key Toyota supplier, which demonstrated the fragility of "Just-in-Time" (JIT) manufacturing systems when faced with supply chain disruptions. Key evidence of this case study includes news coverage of Toyota Motor Corporation's press release confirming the suspension details [59], and automotive industry analysis by [60] regarding the impact on the JIT supply chain.

In February 2022, Toyota Motor Corporation had to suspend operations at all manufacturing plants in Japan. The cause was not a direct attack on Toyota, but a cyberattack on Kojima Industries, a Tier-1 supplier responsible for plastic parts and electronic components [59]. This incident exposed the high dependency risks inherent in lean manufacturing models.

Three attack paths and methods are found in this case. First, by compromising suppliers, attackers breached Kojima Industries' network, reportedly through a remote access mechanism used for third-party communications. Second, the attack disabled Kojima's ability to communicate with Toyota's Kanban (ordering) system, making it impossible to

 

coordinate parts delivery. Third, to prevent the malware from spreading into its own network via the connected supply chain systems, Toyota made the decision to halt all production lines, creating supply chain repercussions.

The impact of the attack highlights how supply chain complexity intertwines with cyberattack risks. On one hand, the suspension affected 28 production lines across 14 plants, resulting in a lost output of approximately 13,000 vehicles in a single day, equivalent to roughly 5% of Toyota's monthly production capacity in Japan [60]. On the other hand, the incident highlighted how a breach at a relatively small, non-famous supplier could paralyze a global automotive giant due to the lack of inventory buffers.

**Key Lessons and Insights (SCCSS Mapping)** The Toyota case illustrates the specific vulnerabilities of "lean" supply chains, where efficiency is prioritized over redundancy.

- Supply Chain Security Subsystem: The extreme reliance on the JIT model meant there was no buffer stock to absorb the disruption. The lack of supply chain redundancy turned a supplier incident into a systemic crisis.

- IT Security Subsystem: The interconnected nature of the Kanban ordering system meant that a failure in the supplier's IT node necessitated a disconnect of the central node (Toyota) to preserve integrity.

- Organizational Security Subsystem: While Toyota's decision to shut down was a prudent containment measure, it revealed a lack of business continuity planning for supplier cyber outages.

In conclusion, the Toyota incident challenges the traditional "efficiency-first" philosophy in the manufacturing sector. It demonstrates that in a digitized supply chain, JIT systems must be paired with "Just-in-Case" cybersecurity contingencies. The event underscores the necessity for manufacturers to assist Tier-1 and Tier-2 suppliers in hardening their defenses, as the resilience of the entire production line is dictated by its most vulnerable component.

### 5.8 Cross-case analysis: Patterns of vulnerability

This section synthesizes findings across the seven landmark cases (Target, Maersk, SolarWinds, Colonial Pipeline, Quanta, MediBank, and Toyota) to identify systemic patterns of failure within the Supply Chain Cyber Security System (SCCSS). Quantitative indicator comparison can be found in Table 5.

The data reveals that while financial loss is the most frequently cited metric, *operational downtime* varies significantly based on the industry's digital-physical coupling. For instance, logistics and energy (Maersk, Colonial) experience multi-day total shutdowns, whereas electronics manufacturing (Quanta) may see no operational halt but suffer catastrophic intellectual property loss. This differentiation confirms that supply chain resilience must be measured by industry-specific degree of impact—whether in production units, recovery days, or data volume—rather than a single financial figure.

For the initial attacks, a universal pattern across all cases is *the exploitation of trust*. In every instance, attackers bypassed the primary target's perimeter by compromising a less secure third-party node. Target (HVAC vendor) and MediBank (IT service desk) demonstrated how non-core vendors with privileged access credentials can become fatal backdoors. SolarWinds and Maersk (NotPetya) revealed that trusted software updates are effective delivery mechanisms for malware, bypassing traditional firewalls that trust signed code. The Toyota case highlighted that in JIT manufacturing, a cyber-physical disconnection at a supplier (Kojima) forces a shutdown of the central node, even without direct infection.

For *internal propagation mechanisms*, once inside, the severity of the impact was consistently determined by the victim's internal architecture and governance/coordination. Maersk and Colonial Pipeline suffered catastrophic operational paralysis because their "flat" network architectures allowed malware to spread unchecked from IT to OT (Operational Technology) or across global offices. Target and SolarWinds illustrated organizational paralysis, where alerts were either ignored due to "alert fatigue" or the dwell time was extended (over a year for SolarWinds) due to a lack of proactive threat hunting.

**Table 5. Quantitative indicators of supply chain cyberattacks.**

| Case | Primary Financial Impact | Operational Downtime/ Recovery | Scale of Data/ Production Loss |
|---|---|---|---|
| Target (2013) | $61M (reported) to $2.2B (est. total) | Response failure despite $1.6M detection tool | 40M credit cards; 70M personal records |
| Maersk (2017) | $250M–$300M (initial); $700M (recovery) | 10 days (reconstruction); 2–3 months (full recovery) | 16,500 servers & 65,000 devices affected |
| SolarWinds (2020) | 11% average loss of annual revenue for victims | 15-month dwell time; 6-month active attack lifecycle | 18,000 + customers affected; 100 high-value targets |
| Colonial (2021) | $4.4M ransom paid ($2.3M recovered) | 5,500 miles of pipeline shut down for 6 days | 100 GB of data stolen; gas prices rose ~4¢/gallon |
| Quanta (2021) | $50M ransom demand (later doubled to $100M) | No "material impact" to operations reported | Leak of proprietary Apple device blueprints |
| MediBank (2022) | $450M (litigation, remediation, and revenue loss) | Targeted systems offline for a short period | 9.7M records; 200 GB of sensitive medical data |
| Toyota (2022) | Lost output worth ~5% of monthly domestic capacity | 14 plants (28 lines) shut down for 1 day | 13,000 vehicles in lost production |

Maersk and Toyota illustrate distinct resilience drivers. Maersk's architectural failure was mitigated by a robust Organizational Subsystem that mobilized effective offline recovery. In contrast, Toyota's Just-in-Time model created a Supply Chain Subsystem fragility, where a lack of buffer stock turned a single supplier breach into a total production halt. Thus, while IT weaknesses trigger incidents, supply chain rigidity determines the operational extent of impact.

The analysis reveals a shift in attacker motives from pure theft to "dual extortion." Quanta Computer and MediBank show that attackers now leverage the sensitive data of downstream clients (e.g., Apple designs, patient diagnoses) to force ransom payments, effectively weaponizing the supply chain relationship itself.

Furthermore, cross-case analysis reveals that information sharing is only as effective as the receiving organization's internal coordination. The SolarWinds incident demonstrates the ideal function of the Supply Chain Subsystem: FireEye's transparent and rapid disclosure of the "Sunburst" indicators allowed the global community to identify and isolate the threat within days, effectively neutralizing the attacker's dwell time advantage. Conversely, the MediBank case illustrates a coordination gap where external intelligence did not trigger internal defense. Despite the availability of threat data, weak internal governance and fragmented decision-making structures delayed the response. Thus, resilience is not defined by the availability of shared intelligence, but by the organizational agility to act upon it.

Finally, comparing Colonial Pipeline and Target illustrates that technical tools are ineffective without organizational enforcement. Colonial's flat architecture forced a total shutdown because it lacked the segmentation to isolate IT from operational systems, whereas Target successfully generated technical alerts that were simply ignored by management. This confirms that "Internal Control" (P2) requires not just technical segmentation, but the organizational authority to act.

Because this study draws on secondary data from publicly documented incidents, the findings are shaped by reporting and visibility biases. Accordingly, the identified propagation mechanisms and governance patterns should be read as robust analytical patterns across the selected cases, but not as fully representative of all supply chain cyberattacks.

### 5.9 Proposition verification and discussion of cross-case variation

The cross-case analysis provides a critical lens through which to evaluate the study's six theoretical propositions, revealing significant boundary conditions and nuances rather than universal confirmation. Regarding Proposition 1, the evidence offers only conditional support for the efficacy of contractual governance. While strict vendor due diligence can mitigate basic third-party risks, contractual mechanisms demonstrate severe boundary limitations when faced with sophisticated

intrusions. The SolarWinds incident illustrates that even mature contractual compliance and service-level agreements cannot adequately audit deeply embedded software updates. This suggests that contractual governance is a necessary but inherently insufficient condition for resilience against advanced persistent threats, as legal obligations cannot substitute for continuous zero-trust technical verification.

Similarly, the evaluation of Proposition 2 reveals that internal governance controls are subject to strict operational and psychological constraints. Strong network segmentation theoretically limits lateral spread, yet the Colonial Pipeline case demonstrates that perceived vulnerability can trigger systemic shutdowns even without confirmed technical spread. Because management lacked absolute confidence in the segmentation between their IT and Operational Technology (OT) networks, they preemptively halted pipeline operations. This indicates that the effectiveness of internal governance is contingent not just on the technical architecture itself, but on the organization's verifiable confidence in that architecture during the fog of a crisis.

When examining relational governance and governance synergy (Propositions 3 and 4), the case data points to a complex dynamic rather than straightforward validation. Proactive information sharing theoretically accelerates joint response, but only when the receiving organization possesses corresponding internal agility. The MediBank incident serves as a critical counter-example where available intelligence and supplier warnings failed to trigger timely internal containment. This partial support highlights a dangerous ambiguity in supply chain governance: participating in advanced threat intelligence sharing without matching internal structural coordination can create a false sense of security while offering negligible practical resilience.

Finally, the dynamics of operational coupling (Proposition 5) and security investment asymmetry (Proposition 6) exhibit notable contextual variations that challenge a universal interpretation. The Toyota shutdown confirms that lean manufacturing drastically accelerates disruption velocity, but the Maersk case reveals that operational decoupling via manual offline recovery is inherently temporary and cannot sustain long-term enterprise viability. Furthermore, while attackers frequently exploit security investment asymmetries to bypass focal firm perimeters, the Quanta Computer breach demonstrates that attackers are increasingly weaponizing this asymmetry rather than just using it as a backdoor. In instances of dual extortion, the relational dependency itself becomes the primary target, complicating the traditional assumption that securing the focal firm's perimeter is the ultimate objective of supply chain cybersecurity.

While the preceding analysis identifies recurring attack vectors and governance failures, understanding systemic resilience requires explaining why cyberattack outcomes differ significantly across the studied cases. Through explicit cross-case comparison, three dimensions of conditional variation emerge: the extent of system-wide collapse, the speed of attack propagation, and the resilience driven by operational coupling.

The first comparative dimension reveals that the extent of system-wide collapse is conditionally dependent on internal governance and architectural segmentation, rather than the initial entry vector. A stark contrast emerges when comparing the Maersk incident with the Target and Colonial Pipeline breaches. At Maersk, the mechanism of Network Flattening led to catastrophic, global operational paralysis because a completely flat network architecture allowed malware to propagate unchecked across all interconnected systems. Conversely, while Target and Colonial Pipeline experienced severe disruptions, their impacts were more contained because internal controls and partial system isolations restricted unrestricted lateral movement. This juxtaposition demonstrates that initial perimeter breaches only escalate into systemic collapse under the specific condition of absent internal structural barriers.

A second critical contrast lies in the speed of attack propagation, which is governed by the intersection of relational governance and internal monitoring capabilities. The NotPetya attack on Maersk exhibited near-simultaneous global propagation, overwhelming systems within minutes. This rapid spread contrasts sharply with the SolarWinds breach, where attackers maintained a stealthy dwell time of over fifteen months. This variation highlights how weak internal monitoring creates Alert Paralysis, allowing threats to linger, whereas proactive information sharing and relational governance—when paired with agile internal response structures—serve as necessary conditions to effectively truncate threat dwell time and

slow lateral movement. Propagation velocity is therefore not merely a technical function of the malware, but a governance-dependent outcome.

Finally, comparing cases with differing degrees of operational coupling illuminates the conditional nature of overall organizational resilience. The physical rigidity of Toyota's Just-in-Time manufacturing model illustrates how tight operational coupling without corresponding digital redundancy amplifies disruption; a single supplier failure forced an immediate, systemic shutdown of twenty-eight production lines. In direct contrast, Maersk mitigated its physical disruption velocity during a total IT collapse by pivoting to offline, manual paper records to maintain baseline port operations. This specific comparison provides strong empirical support for the concept of Synergy Dependency across the SCCSS subsystems. It clarifies that digital resilience is not an isolated IT metric but is conditionally dependent on the structural flexibility of the physical supply chain and the coordinated governance across organizational boundaries.

## 6 Conclusion and recommendations

### 6.1 Theoretical implications

This study offers empirically grounded analytical interpretations that build upon the SCCSS framework [25]. First, the study identifies a synergy dependency between subsystems; our findings suggest that the Supply Chain subsystem's "Relational Governance" (e.g., information sharing) is secondary to the Organizational subsystem's "Internal Governance" (e.g., coordination). This indicates that resilience is not a sum of its parts but a product of hierarchical synergy, where organizational readiness indicates the utility of external intelligence.

Second, the study operationalizes the Points of Penetration (PoPs) by identifying them as the primary sites of cyber-physical conversion. The case evidence—specifically from the Toyota and Maersk incidents—shows that while subsystems define the "domain" of security, the PoPs define the "mechanism" of propagation. This adds a dynamic, process-oriented dimension to the SCCSS architecture. Finally, the research bridges the gap between digital and physical risk management by demonstrating that digital resilience is inextricably linked to the structural design of the physical supply chain. By linking Supply Chain Governance theory with the SCCSS framework, this research provides a basis for future inquiries into the governance-driven propagation of systemic threats [14,41].

Our study identifies that SCCA propagation is not merely a technical spread of malware, but a systemic escalation driven by specific mechanisms across the IT, organizational, and supply chain subsystems. By analyzing the landmark incidents through the SCCSS framework, this research characterizes propagation via the following mechanisms (Table 6):

These mechanisms demonstrate that the trajectory of a supply chain cyberattack is dictated by the structural and relational maturity of the network. For instance, while the Target breach was a failure of Organizational Security (ignoring alerts), the Toyota incident was a failure of Supply Chain Security (rigid JIT dependencies). These insights suggest that effective defense requires focal firms to address not just the "points of entry," but the internal and structural mechanisms that indicate the extent of an inevitable breach.

### 6.2 Managerial implications

The findings indicate that supply chain cyber risk is a systemic managerial issue rather than a purely technical problem. Across the cases, cyber incidents escalated rapidly due to inter-organizational dependencies, highlighting the need for coordinated governance across IT, organizational, and supply chain domains.

First, cybersecurity in supply chains requires active senior management oversight. The scale of disruption observed in multiple cases demonstrates that cyber incidents can quickly evolve into enterprise-level crises, making integration with broader risk management and governance structures essential. Furthermore, the Governance Synergy requirement provides a conceptual foundation for risk managers to evaluate their supplier risk management practices based on an organization's structural coordination agility rather than just perimeter defense compliance.

**Table 6. Summary of propagation mechanisms.**

| Propagation Mechanism and Case Examples | Subsystem Driver | Description of Escalation |
|---|---|---|
| **Network Flattening** (Maersk, Colonial Pipeline) | IT Security | The absence of internal network segmentation and zero-trust protocols allows an attacker to move laterally from a peripheral vendor's access point to core operational systems or global infrastructure. |
| **Alert Paralysis** (Target, SolarWinds) | Organizational Security | Escalation is exacerbated when technical alerts are generated but remain unacted upon due to a lack of defined response procedures, "alert fatigue," or the absence of management authority to initiate containment. |
| **Operational Coupling** (Toyota, Maersk) | Supply Chain Security | In "lean" supply chains, the lack of digital redundancy or inventory buffers (e.g., JIT) ensures that a cyber-physical failure at a single supplier node results in a systemic halt of the entire production line. |
| **Relational Weaponization** (Quanta Computer, MediBank) | Governance Dynamics | Attackers utilize "dual extortion" by stealing the intellectual property or sensitive data of downstream clients, using the supply chain relationship itself as a lever to force ransom payments from the focal firm. |

Second, the results underscore the limitations of static, compliance-oriented vendor assessments. Attacks frequently originated from trusted third parties that formally met security requirements. Managers should therefore adopt risk-based third-party risk management, prioritizing controls based on vendor access, system criticality, and dependency intensity. Firms should proactively support by offering 'security-as-a-service' to peripheral suppliers, thereby hardening the network against the security investment asymmetries that attackers frequently exploit.

Third, internal containment capabilities significantly influence impact severity. Organizations with weak internal segmentation experienced extensive disruption once breached, suggesting that internal access control and segmentation are critical managerial investment priorities for limiting attack propagation.

Finally, while information sharing among supply chain partners supports faster detection and response, its effectiveness depends on clear coordination mechanisms. Shared intelligence must be supported by predefined roles and response procedures to translate awareness into action. These findings offer a blueprint for cross-sector regulatory advocacy (e.g., the U.S. Cybersecurity Information Sharing Act of 2015), encouraging managers to participate in standardized intelligence-sharing structures that ensure external threat data is paired with the internal operational readiness necessary for decisive containment.

Overall, the findings suggest that improving supply chain cyber resilience depends less on isolated security measures and more on coordinated governance and alignment across organizational and supply chain boundaries.

### 6.3 Limitations and future research directions

This study has several limitations that should be acknowledged. First, while this study employs data triangulation to synthesize event timelines, relying entirely on publicly available secondary sources introduces inherent methodological constraints. Each source type carries specific biases: investigative journalism may contain interpretive bias, technical reports often overemphasize technical vulnerabilities at the expense of governance context, and corporate disclosures frequently omit sensitive operational details to manage public relations and legal liability. Although cross-verifying these disparate sources helps construct a more objective narrative, it cannot completely eliminate the blind spots inherent in relying exclusively on external, post-hoc reporting rather than direct organizational access.

Second, as a qualitative multi-case study, the research does not aim for statistical generalization, and causal relationships should be interpreted with caution. This limitation may bias toward observable large-scale failures and underrepresent successfully mitigated incidents.

Third, while this study emphasizes inter-system coordination failures within the SCCSS framework, alternative explanations such as firm size, national culture, industry-specific technological constraints, and the presence of state-sponsored threat actors may also influence attack propagation and should be considered when interpreting the findings.

Future research could address these limitations in several ways. Quantitative studies may examine how specific governance mechanisms influence the diffusion and impact of cyber risk across supply chain networks. Longitudinal research could further explore how organizations adapt governance structures over time in response to repeated cyber incidents. Additionally, greater attention to behavioral and organizational factors, such as security culture and inter-organizational coordination practices, would deepen understanding of how governance frameworks are implemented in practice. Expanding analysis to emerging digital contexts, including AI-driven supply chains, also represents a promising direction for future inquiry.

## Supporting information

**S1 File. Supporting information.**
(DOCX)

## Acknowledgements

The authors certify that generative AI tools were not used to generate data, conduct analysis, or derive conclusions. Google Gemini was employed exclusively for copy-editing to enhance the manuscript's readability and flow. The authors have reviewed all language adjustments and maintain full responsibility for the accuracy, integrity, and originality of the work. The authors thank the academic editor and four reviewers for their helpful comments, which have greatly improved this paper.

## Author contributions

**Conceptualization:** Xiaoya Ba.

**Formal analysis:** Xue Zhang, Bo Li.

**Methodology:** Xue Zhang, Xiaoya Ba.

**Supervision:** Bo Li.

**Validation:** Xue Zhang.

**Visualization:** Xiaoya Ba.

**Writing – original draft:** Xiaoya Ba.

**Writing – review & editing:** Xue Zhang, Bo Li.

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
