## [Decision Letter · Decision Letter 0]

16 Mar 2026

PONE-D-25-68972Cyberattacks in Supply Chains: A Multi-Case StudyPLOS One

Dear Dr. Li,

Thank you for submitting your manuscript to PLOS ONE. After careful consideration, we feel that it has merit but does not fully meet PLOS ONE’s publication criteria as it currently stands. Therefore, we invite you to submit a revised version of the manuscript that addresses the points raised during the review process.

We look forward to receiving your revised manuscript.

Kind regards,

Eugenio Oropallo, Ph.D., Eng.

Academic Editor

PLOS One

Journal Requirements:

“The authors received no specific funding for this work.”

3. Please note that your Data Availability Statement is currently missing the repository name and/or the DOI/accession number of each dataset OR a direct link to access each database. If your manuscript is accepted for publication, you will be asked to provide these details on a very short timeline. We therefore suggest that you provide this information now, though we will not hold up the peer review process if you are unable.

4. We note you have included a table to which you do not refer in the text of your manuscript. Please ensure that you refer to Table 3 in your text; if accepted, production will need this reference to link the reader to the Table.

Additional Editor Comments:

The manuscript investigates the dynamics of Supply Chain Cyberattacks (SCCAs) through a qualitative multi-case study of seven major cyber incidents.

The authors apply the SCCSS framework, which conceptualizes cybersecurity across three interconnected subsystems:

• IT security

• organizational security

• supply chain security.

Using secondary data sources such as government reports, forensic analyses, corporate disclosures, and investigative journalism, the study maps attack vectors and propagation mechanisms across these subsystems.

The findings suggest that:

• most cyberattacks originate from third-party connections within supply chains,

• attack propagation is often amplified by coordination failures across systems,

• resilience improves when organizations implement information sharing, internal controls, and coordinated governance mechanisms.

The paper concludes that effective cybersecurity governance requires a system-level perspective rather than isolated defensive measures.

The topic is highly relevant given the increasing importance of supply chain cybersecurity. The manuscript offers a useful synthesis of several prominent incidents. However, the current version presents important limitations regarding theoretical positioning, methodological transparency, and analytical depth.

The manuscript claims to extend the SCCSS framework and contribute to the understanding of supply chain cyber risk. However, the theoretical contribution remains unclear.

For instance, the introduction states that the study aims to examine: “how system coordination and information sharing enhance resilience by establishing a system-level defense.”

However, the paper mainly applies the SCCSS framework rather than substantially extending it. The analysis largely maps observed events to existing categories (IT, organizational, supply chain security).

Consequently, the manuscript would benefit from clarifying:

• what specific theoretical advancement is achieved,

• whether the study tests, refines, or extends the SCCSS framework,

• how the findings advance existing literature on cyber resilience or supply chain risk governance.

Without this clarification, the contribution risks appearing primarily descriptive.

The methodology section describes a qualitative multi-case study based on seven well-known cyber incidents. While the authors mention that cases were selected to represent diverse sectors and attack types, the selection criteria remain insufficiently justified.

For example, the paper states: “Selection criteria required that each case involve a cyberattack targeting these subsystems, offer rich verifiable data, and reflect diverse sectors and motives.”

However, the manuscript does not explain:

• how many candidate cases were initially considered,

• whether any systematic sampling strategy was used,

• whether the cases represent theoretical replication or convenience sampling.

Since the study focuses on high-profile incidents, there is a risk of selection bias, which should be explicitly discussed.

The empirical analysis relies entirely on publicly available secondary sources, including:

• investigative journalism

• corporate reports

• government documents

• forensic analyses.

While triangulation is mentioned, the manuscript does not fully discuss the limitations associated with this data source.

For example:

• journalistic reports may contain interpretive bias,

• technical reports may focus primarily on technical vulnerabilities rather than governance issues,

• corporate disclosures may omit sensitive details.

Although the authors acknowledge this limitation in the discussion, a deeper methodological reflection is necessary.

The manuscript presents detailed narrative descriptions of several incidents. For instance, the Quanta Computer case illustrates how attackers exploited supply chain relationships to access sensitive intellectual property.

Similarly, the MediBank case describes credential theft and large-scale data exfiltration resulting from weak access controls.

While these descriptions are informative, the analysis often remains descriptive rather than analytical.

In many sections the paper primarily recounts the events and then maps them to SCCSS categories. Stronger analytical insights could be generated by:

• systematically comparing cases,

• identifying patterns across incidents,

• explaining why some attacks propagate more severely than others.

The manuscript states that cross-case synthesis was conducted to extract higher-level insights. However, the results section does not clearly present:

• structured cross-case comparisons,

• tables summarizing key variables,

• systematic pattern identification.

A comparative table summarizing the seven cases (e.g., entry vector, propagation mechanism, governance failure, mitigation response) would significantly improve clarity.

While the manuscript refers to several conceptual perspectives (e.g., supply chain governance theory, resilience and robustness concepts), the theoretical grounding of the study remains relatively limited. The analysis is primarily structured around the SCCSS framework, which functions more as an analytical classification model than as a fully developed theoretical lens. As a result, the manuscript would benefit from a stronger integration of established theories from supply chain management, organizational governance, or systems theory to better explain the mechanisms underlying cyberattack propagation and resilience. Furthermore the SCCSS framework looks to be used primarly as coding tool and not as a theory building tool.

The manuscript does not clearly articulate explicit research questions. Explicitly stating the research questions would improve the structure of the study.

Reviewers' comments:

Reviewer's Responses to Questions

**Comments to the Author**

1. Is the manuscript technically sound, and do the data support the conclusions?

Reviewer #1: Yes

Reviewer #2: Partly

Reviewer #3: Partly

Reviewer #4: Partly

2. Has the statistical analysis been performed appropriately and rigorously? 

Reviewer #1: Yes

Reviewer #2: Yes

Reviewer #3: Yes

Reviewer #4: N/A

3. Have the authors made all data underlying the findings in their manuscript fully available?

Reviewer #1: Yes

Reviewer #2: Yes

Reviewer #3: Yes

Reviewer #4: Yes

4. Is the manuscript presented in an intelligible fashion and written in standard English?

Reviewer #1: Yes

Reviewer #2: Yes

Reviewer #3: No

Reviewer #4: Yes

5. Review Comments to the Author

Reviewer #1: Although the manuscript addresses an important topic (supply chain cyberattacks), the conceptual novelty is limited. The study primarily applies the existing SCCSS framework to well-known public cyber incidents, without developing new theoretical constructs or analytical mechanisms.

All case materials are derived from secondary public sources (media reports, government documents, and technical blogs). No primary data (e.g., interviews, internal reports, or practitioner surveys) are used, which weakens empirical depth and originality.

Case selection criteria require further clarification. The manuscript focuses exclusively on high-profile incidents (e.g., Target, SolarWinds, Maersk), which may introduce selection bias and limit generalizability to typical supply chain organizations.

The SCCSS framework is adopted rather than substantially extended. The propositions (P1–P4) largely restate established concepts such as governance maturity, internal control, and information sharing, providing limited theoretical advancement.

The causal relationships between governance mechanisms and cyber resilience remain largely inferential. The analysis does not rigorously distinguish correlation from causation across the multi-case comparisons.

While qualitative coding is described, inter-coder reliability or independent validation procedures are not reported, raising concerns regarding analytical subjectivity.

Several sections rely heavily on narrative description of incidents, with limited analytical abstraction beyond mapping events to SCCSS categories.

Quantitative indicators (e.g., recovery time, financial loss, operational downtime) are inconsistently reported across cases, making cross-case comparison less systematic.

The discussion emphasizes well-known cybersecurity practices (Zero Trust, MFA, vendor audits), but offers limited actionable differentiation from existing cybersecurity guidelines (e.g., NIST SP 800-161).

The manuscript would benefit from a clearer visual synthesis (e.g., conceptual diagram or summary table) illustrating cross-case patterns of vulnerability and resilience.

Some references rely on technical blogs or news outlets rather than peer-reviewed literature, which weakens academic rigor.

The conclusion largely reiterates earlier findings without sufficiently articulating broader theoretical implications for supply chain governance research.

Language in several sections is managerial or descriptive rather than analytical, and could be tightened to improve academic tone.

Reviewer #2: Summary of the research and overall impression

This manuscript does a good job of examining supply chain cyberattacks using a qualitative multi-case study approach. The authors analyze seven real-world cyberattack incidents across multiple industries and apply the Supply Chain Cyber Security System (SCCSS) framework to identify common attack vectors, propagation mechanisms, and governance failures. The study aims to understand how weaknesses in IT systems, organizational practices, and supply chain governance contribute to cyberattack entry and spread.

The manuscript addresses a relevant and timely research problem and uses an appropriate qualitative methodology to achieve this. It is kind of a tedious read, but a good one ultimately.

Overall, the study is technically sound and contributes useful insights into supply chain cybersecurity governance. However, several minor issues should be addressed to improve methodological clarity and ensure full compliance with journal formatting standards, and these will be addressed subsequently.

Major Issues

1. Reference numbering and formatting should be corrected

Unless I am missing something, and please let me know if I am, but References should be numbered strictly in order of first appearance. For example, early in the Introduction, references are cited out of sequence, which suggests that numbering was not properly synchronized. These Numbering inconsistencies should be addressed in full compliance with PLOS ONE reference style (Vancouver).

2. Clarification of coding reliability and consistency

The manuscript describes the coding process used to enhance reliability, which strengthens the analysis. However, the manuscript does not clearly explain whether multiple researchers independently coded the data or how coding disagreements were resolved. The authors should provide additional clarification on how coding consistency and reliability were ensured. This is the chief reason for the "partly" answer in question 1.

minor issues

1. Relying fully on secondary sources (such as reports and news articles) can be tricky. While this is acceptable for case studies, it limits the strength of the conclusions.

2. The manuscript is generally well written and easy to understand, but there are minor grammatical issues and occasional repetitive phrasing. These can be corrected during revision.

Conclusion

This is a technically sound paper, and analysis has been done well for the most part. It, however requires revisions before it is quite ready for publication.

Reviewer #3: In the abstract and conclusion, the contribution of this paper is not well presented. In the conclusion and abstract, highlight the novelty of the paper.

The introduction is weak and should include the research question, the aim of the paper and the contribution.

In related work…. Many researches work on this idea. What is really the novelty as compared to other studies? What is the new and the difference between the previous works and present work?.

Improve the quality of literature along with the latest literature.

The explanation of the related work needs to be criticized and improved in general.

What about last updating in this topic and new references from 2019-2025? The survey of existing literature is not sufficient. It would useful to include in the Introduction of the paper some discussion on other possible real applications of the obtained results.

Weak conclusion

Conclusion should be more specific with improvement writing quality.

A suggestion for future work should be added in the conclusion section.

- Rewrite the references according to journal template.

-Please strictly follow the instructions to the format specified in the journal template for preparing the paper

The format and English writing of this paper should be improved. The paper needs language revision.

Reviewer #4: This study examines Supply Chain Cyberattacks (SCCAs) through a qualitative multi-case analysis of seven major cyber incidents across industries such as retail, logistics, energy, and healthcare. Using the Supply Chain Cyber Security System (SCCSS) framework, the authors analyze how vulnerabilities in IT systems, organizational governance, and supply chain relationships contribute to cyberattack entry points and propagation. The findings show that most attacks originate from trusted third-party connections, and their operational impact is amplified by weak internal controls and poor coordination across systems. The study concludes that improving cyber resilience requires cross-organizational governance mechanisms, including stronger third-party risk management, network segmentation, and proactive information sharing among supply chain partners.

The introduction concludes by stating that the study investigates the propagation of supply chain cyberattacks (SCCAs). However, the research questions span several distinct phenomena, including attack entry points (RQ1), inter-system escalation (RQ2), and recovery through intelligence sharing (RQ3). Only RQ2 directly addresses propagation. The manuscript would benefit from clarifying the conceptual relationship among these questions and explaining how they collectively contribute to understanding cyberattack propagation within the SCCSS framework.

Throughout the manuscript, the study is framed as investigating the propagation of supply chain cyberattacks (SCCAs). However, the conclusion does not clearly articulate what specific insights the study offers regarding propagation mechanisms beyond the observation that attacks often originate from third-party entry points and spread through poorly coordinated systems. The authors could clarify the concept of propagation by specifying the mechanisms through which cyber incidents escalate across systems. Explicitly summarizing these mechanisms in the conclusion would help consolidate the study’s contribution and provide a clearer conceptual understanding of how SCCAs spread across supply chain networks.

The paper introduces Points of Integration (technology, human resources, and physical processes) as mechanisms connecting the three SCCSS subsystems. However, the relationship between these PoPs and the subsystem architecture in Table 1 is not clearly articulated. It remains unclear how the PoPs operationalize or interact with the IT, organizational, and supply chain security subsystems. Please consider providing an explicit mapping or conceptual diagram that would substantially improve the clarity of the framework.

It is unclear whether the selected cases represent typical supply chain cyberattacks or simply the most visible ones. Without systematic selection criteria, the findings may reflect selection bias rather than generalizable patterns. The authors could strengthen methodological transparency by explicitly describing the case selection criteria and explaining why these seven cases provide appropriate analytical coverage of supply chain cyber risks. For example, the authors might clarify whether cases were selected to represent variation in industries, attack vectors, governance failures, or supply chain structures. Providing a short table summarizing selection rationale or linking case characteristics to the research questions would improve confidence in the robustness of the case design.

The manuscript states that deductive thematic coding was conducted using the SCCSS framework to classify empirical observations into IT, organizational, and supply chain subsystems. However, the description of the coding process remains relatively brief and does not fully clarify how the analysis was conducted in practice. It is unclear whether multiple researchers participated in the coding process, how coding consistency was ensured, or whether any form of intercoder agreement assessment was performed. Without additional detail, it is difficult for readers to assess the reliability and replicability of the qualitative analysis. The authors could enhance methodological rigor by providing more detail on the coding procedure. For instance, clarifying how many researchers participated in coding, whether the coding scheme was iteratively refined, and how disagreements were resolved would strengthen transparency. If applicable, reporting an intercoder agreement measure or describing a consensus-based coding process would further reinforce the credibility of the analytical procedure. Even a brief appendix illustrating sample coded excerpts could significantly improve reproducibility.

Several core concepts: inter-system coordination, attack propagation, governance robustness, and recovery speed, play a central role in the cross-case analysis. However, these constructs are primarily discussed in descriptive terms, and their operationalization across cases is not always clearly specified. As a result, it becomes difficult to determine how these concepts were consistently assessed across the seven incidents or how the authors compared cases. To improve analytical clarity, the authors could more explicitly define how key constructs were operationalized in the case analysis. For example, indicators such as attack entry vector, internal propagation mechanism, detection delay, or recovery duration could be consistently mapped across cases. A summary table that operationalizes these constructs and shows how they were observed in each case would help readers understand how the cross-case comparisons were conducted and would strengthen the overall methodological transparency of the study.

The manuscript claims that it “empirically validates the Supply Chain Cyber Security System (SCCSS) framework” and makes a “distinct theoretical contribution” by demonstrating that cyber risk is a systemic property emerging from the interdependence of IT, organizational, and supply chain subsystems. However, the empirical analysis primarily illustrates how well-known cyber incidents can be interpreted through the SCCSS lens, rather than demonstrating clear theoretical advancement. The cases show that vulnerabilities may arise from IT weaknesses, organizational failures, and supply chain relationships, those insights are broadly consistent with existing cybersecurity and supply chain risk management literature. The study appears to apply an existing framework to multiple cases, but it is less clear how the findings extend, refine, or challenge the framework in a theoretically meaningful way. The authors should strengthen the theoretical contribution by explicitly articulating how the case evidence refines the framework, not simply confirming it.

The manuscript addresses an important and timely topic and provides insightful case analyses of major supply chain cyber incidents. However, several aspects of the methodological design and theoretical positioning require further clarification. In particular, the case selection rationale, transparency of the coding procedure, and operationalization of key constructs should be strengthened. Additionally, the manuscript would benefit from a more cautious articulation of its theoretical contribution. I wish the authors the best in revising the manuscript and look forward to seeing how the study develops in the next iteration.

6. PLOS authors have the option to publish the peer review history of their article (what does this mean?). If published, this will include your full peer review and any attached files.

Reviewer #1: **Yes:** Wenhao Ren

Reviewer #2: No

Reviewer #3: No

Reviewer #4: No

---

## [Author Response · Author response to Decision Letter 1]

6 Apr 2026

Please see the response letter. Thank you very much!

---

## [Decision Letter · Decision Letter 1]

27 Apr 2026

PONE-D-25-68972R1Cyberattacks in Supply Chains: A Multi-Case StudyPLOS One

Dear Dr. Li,

Thank you for submitting your manuscript to PLOS ONE. After careful consideration, we feel that it has merit but does not fully meet PLOS ONE’s publication criteria as it currently stands. Therefore, we invite you to submit a revised version of the manuscript that addresses the points raised during the review process.

We look forward to receiving your revised manuscript.

Kind regards,

Eugenio Oropallo, Ph.D., Eng.

Academic Editor

PLOS One

Journal Requirements:

Additional Editor Comments:

Dear Authors,

Thank you for submitting your revised manuscript. The manuscript has now been evaluated by the reviewers and assessed editorially.

While the reviewers have expressed generally positive views regarding the relevance of the topic and the clarity of the manuscript, I find that the paper, in its current form, requires a few more revisions before it can be considered for publication. I am therefore inviting you to submit a major revision.

Your study addresses an important and timely issue (cyber risk propagation in supply chains) and presents a well-structured qualitative multi-case analysis of seven high-profile incidents. The manuscript is clearly written, logically organised, and grounded in relevant literature. The use of the SCCSS framework to structure the analysis is appropriate, and the comparative perspective across cases has the potential to offer valuable insights.

However, several concerns must be addressed to ensure that the manuscript meets the standards of methodological rigor and interpretative balance required for publication.

Alignment between claims and evidence:

The manuscript presents strong theoretical claims, however, these contributions are not sufficiently substantiated by the analysis. The current evidence supports the identification of recurring patterns across cases but does not justify the level of theoretical advancement claimed. I encourage you to recalibrate the framing of your contributions, positioning them more cautiously as empirically grounded insights or analytical interpretations rather than formal theoretical extensions...or to stress the discussion and the result analysis in order to allign them with your theoretical claims.

Section 5.9 (“Proposition Verification”) reads primarily as a confirmatory narrative in which all propositions are reported as “aligned.” This approach does not provide a sufficiently critical or analytical evaluation. Please revise this section to engage more critically with the evidence, acknowledge ambiguity, boundary conditions, or partial support, avoid presenting propositions as universally confirmed.

While the manuscript includes a cross-case synthesis, the comparative analysis remains largely descriptive. I encourage you to deepen this section by more explicitly contrasting cases, highlighting both differences and similarities, and clarifying under what conditions particular mechanisms or outcomes emerge.

Although the manuscript acknowledges the use of secondary data and potential biases, these limitations should be more explicitly integrated into the interpretation of findings.

Some of the managerial and policy implications (e.g., applications to insurance models, digital twin simulations, and regulatory frameworks) extend beyond what is directly supported by the empirical analysis. Please ensure that all implications are clearly grounded in the presented evidence, and consider moderating or removing those that are more speculative in nature.

In summary, the manuscript has clear potential but requires revisions to ensure that its claims are appropriately supported and that the analysis meets the expected level of rigour. I would be happy to consider a revised version that addresses the points outlined above.

In order to help you in your last effort and to better align the manuscript to the journal, I list some references useful for your work:

- 10.1111/itor.70072

- 10.1371/journal.pone.0335128

- 10.1016/j.jik.2026.100939

- 10.1371/journal.pone.0344098

- 10.1109/TEM.2025.3648054

- 10.3390/systems14020132

Please provide a detailed, point-by-point response to all editorial comments when resubmitting your manuscript.

Reviewers' comments:

Reviewer's Responses to Questions

**Comments to the Author**

1. If the authors have adequately addressed your comments raised in a previous round of review and you feel that this manuscript is now acceptable for publication, you may indicate that here to bypass the “Comments to the Author” section, enter your conflict of interest statement in the “Confidential to Editor” section, and submit your "Accept" recommendation.

Reviewer #1: All comments have been addressed

Reviewer #2: All comments have been addressed

Reviewer #3: All comments have been addressed

Reviewer #4: All comments have been addressed

2. Is the manuscript technically sound, and do the data support the conclusions?

Reviewer #1: Yes

Reviewer #2: Yes

Reviewer #3: Yes

Reviewer #4: Yes

3. Has the statistical analysis been performed appropriately and rigorously? 

Reviewer #1: Yes

Reviewer #2: Yes

Reviewer #3: Yes

Reviewer #4: Yes

4. Have the authors made all data underlying the findings in their manuscript fully available?

Reviewer #1: Yes

Reviewer #2: Yes

Reviewer #3: Yes

Reviewer #4: Yes

5. Is the manuscript presented in an intelligible fashion and written in standard English?

Reviewer #1: Yes

Reviewer #2: Yes

Reviewer #3: Yes

Reviewer #4: Yes

6. Review Comments to the Author

Reviewer #1: The author responded to my question and suggested that the paper be accepted for publication immediately.

Reviewer #2: (No Response)

Reviewer #3: Cyberattacks in supply chains: a multi-case study

All comments have been addressed

accepted without modification

Reviewer #4: I appreciate the authors’ careful and thoughtful revisions in response to my comments. Overall, I find that the authors have made substantial improvements to the manuscript. Good luck!

7. PLOS authors have the option to publish the peer review history of their article (what does this mean?). If published, this will include your full peer review and any attached files.

Reviewer #1: No

Reviewer #2: No

Reviewer #3: No

Reviewer #4: No

To ensure your figures meet our technical requirements, please review our figure guidelines: https://journals.plos.org/plosone/s/figure

---

## [Editor Report · Decision Letter 2]

8 May 2026

Cyberattacks in Supply Chains: A Multi-Case Study

PONE-D-25-68972R2

Dear Dr. Li,

We’re pleased to inform you that your manuscript has been judged scientifically suitable for publication and will be formally accepted for publication once it meets all outstanding technical requirements.

Kind regards,

Eugenio Oropallo, Ph.D., Eng.

Academic Editor

PLOS One

---

## [Editor Report · Acceptance letter]

PONE-D-25-68972R2

PLOS One

Dear Dr. Li,

I'm pleased to inform you that your manuscript has been deemed suitable for publication in PLOS One. Congratulations! Your manuscript is now being handed over to our production team.

Kind regards,

on behalf of

Dr. Eugenio Oropallo

Academic Editor

PLOS One